# Mouse Tumor Models for Advanced Cancer Immunotherapy

**DOI:** 10.3390/ijms21114118

**Published:** 2020-06-09

**Authors:** Daria S. Chulpanova, Kristina V. Kitaeva, Catrin S. Rutland, Albert A. Rizvanov, Valeriya V. Solovyeva

**Affiliations:** 1Institute of Fundamental Medicine and Biology, Kazan Federal University, 420008 Kazan, Russia; daryachulpanova@gmail.com (D.S.C.); olleth@mail.ru (K.V.K.); rizvanov@gmail.com (A.A.R.); 2Faculty of Medicine and Health Sciences, University of Medicine, Nottingham NG7 2HA, UK; Catrin.Rutland@nottingham.ac.uk

**Keywords:** cancer mouse models, cancer immunotherapy, immune checkpoint inhibitors, CAR T-cell therapy

## Abstract

Recent advances in the development of new methods of cancer immunotherapy require the production of complex cancer animal models that reliably reflect the complexity of the tumor and its microenvironment. Mice are good animals to create tumor models because they are low cost, have a short reproductive cycle, exhibit high tumor growth rates, and can be easily genetically modified. However, the obvious problem of these models is the high failure rate observed in human clinical trials after promising results obtained in mouse models. In order to increase the reliability of the results obtained in mice, the tumor model should reflect the heterogeneity of the tumor, contain components of the tumor microenvironment, in particular immune cells, to which the action of immunotherapeutic drugs are directed. This review discusses the current immunocompetent and immunocompromised mouse models of human tumors that are used to evaluate the effectiveness of immunotherapeutic agents, in particular chimeric antigen receptor (CAR) T-cells and immune checkpoint inhibitors.

## 1. Introduction

The tumor microenvironment (TME) consists of a large number of different normal cells that play an important role in the development and progression of the tumor [1]. Stromal fibroblasts, infiltrating immune cells, blood and lymphatic vessels, the extracellular matrix, as well as cytokines and growth factors secreted by TME cells, can contribute both positively and negatively to tumor development [2]. Immunotherapy targets the immune component of TME in order to enhance the antitumor immune response or overcome the ability of the tumor to avoid immunological surveillance [3]. Recent advances in understanding interactions between the tumor and the immune system have allowed the development of new immunotherapy. Approaches such as immune checkpoint inhibitors (ICIs) and chimeric antigen receptor (CAR) T-cells, aim to use the immune system to create effective antitumor responses [4,5]. ICIs are a new class of antitumor immunotherapeutic agents, which act as suppressors of many immune checkpoints, especially in cytotoxic T-cells [6]. Under normal conditions, immune checkpoints can suppress the excessive immune response activation, the development of autoimmune reactions and the damage of the body’s own healthy tissue [7]. However, the expression of some immune checkpoint ligands on the tumor cell surface allows the tumor to suppress the antitumor immune response and to evade the immune surveillance [8]. ICIs can disrupt these inhibitory signals and restore T-cell cytotoxicity against tumor cells [9]. CAR T-cell-based therapy involves the genetic modification of a patient’s autologous T-cells to express CAR targeted at a tumor-specific antigen [10]. Expression of tumor-specific TCR allows CAR T-cells to directly identify the tumor antigen independently of major histocompatibility complex (MHC) and overcome some of the main mechanisms by which tumors can prevent MHC-based detection by T-cells [11,12]. The binding of CAR T-cells with cancer cells stimulates T-cell activation, proliferation and cytotoxic effect against the tumor [13].

To evaluate the effectiveness of immunotherapy, animal models are required in which a human tumor and its microenvironment are genetically, physiologically and anatomically modeled so that the formation and development of the tumor in humans are reliably reflected. Mice are low in cost relative to other animal models, have a short reproductive cycle, exhibit high tumor growth rate, and can be easily genetically modified [14]. These advantages make mouse models a good tool to evaluate the effectiveness of immunotherapeutic approaches for the cancer treatment. However, the ability to translate encouraging results of immunotherapy from bench to clinic is challenging now because of the high failure rate observed in human clinical trials after promising results obtained in mouse models. This review discusses the current immunocompetent and immunocompromised mouse models of human tumors that are used to evaluate the effectiveness of immunotherapeutic agents, in particular CAR T-cells and immune checkpoint inhibitors.

## 2. Syngeneic Tumor Models

Transplantation of in vitro cultured tumor cells into immunocompetent mice is the oldest and most commonly utilized approach to investigate antitumor therapy, including immunotherapy (Figure 1A) [15]. Spontaneous, carcinogenic or transgenic tumor cell lines can be transplanted into such mouse strains as C57BL/6, BALB/c and FVB [16,17,18,19,20]. The creation of such models takes a short period of time, since transplanted subcutaneously or intravenously cells grow in the animal within a few weeks [21]. The fast kinetics of tumor growth in syngeneic models often provides an inadequate time interval for evaluating the effectiveness of immunotherapy, since usually the effect of immunomodulating treatment develops gradually and is estimated by increasing survival rate [15]. Also, in syngeneic models it is impossible to evaluate the effectiveness of immunotherapeutic drugs at earlier stages of the tumor development [22].

Undoubtedly, the formation of transplanted tumors is different from tumors originating de novo. Moreover, the tumor microenvironment will be mostly determined by the local innate immune response triggered by injection-induced inflammation and the presence of a large number of tumor cells [23,24]. Usually, tumor cells are injected subcutaneously, as it is easier to track tumor development [25]. In order to make TME closer to that of people, the tumor cells can be orthotopically transplanted into the corresponding organs [26], such as intravenous administration of leukemia and lymphoma cells [27], injection of breast tumor cell lines into mammary adipose tissue [28], intrapancreatic injection of pancreatic duct adenocarcinoma [29], and intracranial injection for glioblastoma cell lines [30,31]. These administration routes more accurately reflect TME, but require more complicated manipulations and special equipment for both the transplantation and monitoring of the further tumor development [32].

Besides the fact that TME may prove to be unnaturally homogeneous, the syngeneic model of the tumor also lacks genomic heterogeneity, which makes each tumor unique [14]. The absence of mutational heterogeneity in syngeneic tumors is partly due to the absence of cancer stem cells and other populations of progenitor cells that are presented in TME and provide a constant source of new mutations for tumor evolution [33]. In addition, cells intended for transplantation often undergo adaptation to harsh in vitro or in vivo conditions, which leads to a decrease in the heterogeneity of tumor cells [34].

To study the effectiveness of CAR T-cell therapy in syngeneic models, mouse cell-derived CAR-T-cells are used [35]. Mouse models allow the toxicity of CAR T-cells to be determined, which depends on both the dose and the presence of co-stimulatory domains in the chimeric receptor [36]. A syngeneic model of lymphoma in BALB/c mice originally showed that first-generation CAR T-cells (without co-stimulatory domains) killed lymphoma cells, but did not cause side effects [37]. In contrast, second generation CAR T-cells with CD28 co-stimulatory domain induced B-cell aplasia and chronic toxicity, accompanied with an increase in the number of suppressor cells. It is worth noting the toxicity described above has not been observed in lymphoma models created in C3H or C57BL/6 mouse strains, which means the results vary depending on the mouse strain [38].

Syngeneic tumor models are also used to investigate the antitumor activity of ICIs, including anti-cytotoxic T lymphocyte-associated antigen-4 (CTLA-4) [39] and anti-programmed death (PD)-1 anti-PD-L1 antibodies [31,40]. For example, using the syngeneic model of immunocompetent B6 mice transplanted with E.G7 hematopoietic cell line or its analogue with PD-L1 deficiency it has been shown that the PD-L1 pathway blockade contributed to the rejection of tumor cells in mice transplanted with both wild-type (WT) E.G7 cells or PD-L1 deficient E.G7 cells in the same degree. Thus, the expression of PD-L1 on the cells of TME (either tumor infiltrating leukocytes or stromal cells) may contribute more significantly than the expression of PD-L1 on tumor cells [41]. One of the significant drawbacks of using syngeneic models to evaluate the effectiveness of ICIs is that the rapid growth of the tumors in syngeneic mouse models does not contribute towards the development of the chronic inflammatory environment typical for human tumors. Whilst during tumor formation, immunological inhibitory pathways associated with inflammation, such as the PD-1/PD-L1 axis, are activated, which contributes significantly to TME remodeling [42].

## 3. Genetically Engineered Mouse Models

The next step in the development of mouse tumor models was the creation of the mice in which autochthonous tumors develop in specific tissue due to the inclusion of specific changes in the genome (Figure 1B). Genetically engineered mouse models (GEMMs) are usually produced using transgenic technologies to provide systemic or tissue-specific expression of oncogenes, such as *KRAS* and *MYC* in breast cancer [43] and/or deletion of cancer suppressor genes, such as *PTEN* and *TP53* in prostate cancer [44]. These transgenic models can be further divided into germline GEMMs and non-germline GEMMs [45]. Germline GEMMs have mutations that lead to the spontaneous development of malignant neoplasms. For example, it has been shown that in mice with a *TP53* gene mutation, a wide range of solid and hematological malignancies develops [46]. Germline GEMMs have allowed the detailed study of the mechanisms of tumor formation and development, but it is very labor-intensive and does not allow control over the moment and place of tumor onset [47]. Non-germline GEMMs, on the other hand, provide spatiotemporal control of the onset of transformation. Induction of somatic mutations at a selected time and in a specific tissue can be achieved using various systems, for example, the tamoxifen-inducible Cre-loxP system in which, after the endogenous activation of Cre-recombinase by tamoxifen, any gene flanked by loxP recombination sites is deleted [48]. Also, RNA interference (RNAi) using short hairpin RNAs (shRNAs) are used to create non-germline models. For example, shRNA-mediated suppression of adenomatous polyposis coli (APC) tumor suppressor in the presence of *KRAS* and *TRP53* mutations induces the development of colon carcinomas that undergo stable regression after the restoration of APC expression by disabling shRNA expression [49]. The CRISPR/Cas9 technology has recently been actively used for somatic editing of oncogenes, due to which models of hepatocellular carcinoma [50], lung cancer [51], breast cancer [52] have been created. Although the CRISPR/Cas9 gene editing system is extremely efficient in vivo, somatic Cas9 delivery can trigger Cas9-specific immune responses, which will lead to the elimination of cells expressing Cas9 [53]. Moreover, CRISPR/Cas9-mediated genome editing can create unwanted mutations outside the target [54].

In general, de novo tumor formation provides organization of complex TME as well as allowing the tumor to undergo immune tolerance, immuno-editing and/or immunosuppressive processes [55]. Sequential tumor development is a critical advantage of GEMMs compared to syngeneic tumor models, making them especially important for evaluating immunotherapy methods. Also, the interaction of the tumor with the immune system leads to the formation of heterogeneity, which can be enhanced by affecting the genes associated with mismatch repair and genomic stability, such as *MLH1* [56] and *BRCA1/2* [57]. An increase in the mutational burden can lead to the formation of neoantigens that can be recognized by immune cells [58]. However, this can lead to the evolution of the anti-cancer immune response, which will affect the effectiveness of immunotherapy [59]. In view of this, the GEMMs phenotype is poorly reproducible in comparison with syngeneic models. Another challenge is the requirement for non-invasive imaging techniques, such as ultrasound or magnetic resonance imaging, to monitor tumor development and evaluate antitumor immune responses [60].

To study the effectiveness of CAR T-cells, genetically modified mice are infrequently-accessed compare to syngeneic or patient-derived xenograft models. Most often, mice are genetically modified to express human tumor associated antigen (TAA) transgenes (the mouse TAAs are knocked out) and the tumor is syngeneic [61]. Murine T-cells that express human TAA are used in these studies [62,63]. Since most TAAs are expressed not only in tumors, but also at lower levels in healthy tissues, transgenic mice serve as an important model for evaluating the undesirable side effects that are observed in CAR T-cell therapy [64]. For example, C57BL/6 mice were genetically modified to express human carcinoembryonic antigen (CEA), which is overexpressed in many cancers of the gastrointestinal and pulmonary tracts, but it is not tumor specific and is also expressed in healthy intestine and lung tissue. Transgenic mice were treated with anti-CEA CAR T-cells, which resulted in long term tumor eradication, but also led to the heavy infiltration of the intestines and lung with anti-CEA CAR T-cells in the transgenic but not in WT mice. However, despite the strong infiltration of healthy tissues, autoimmune inflammatory responses were not observed [61].

In the field of ICIs, genetically modified models are also not often used. A pancreatic ductal adenocarcinoma model, created using Cre-Lox technology, which activates mutant endogenous alleles of the *KRAS* and *TRP53* genes, has been described [65]. Unfortunately, anti-PD-1 antibody therapy did not lead to a significant effect on tumor growth in these mouse models [66].

## 4. Carcinogen-Induced Tumor Models

Tumor formation in mice can also be induced by carcinogens (CI) (Figure 1C) [67,68]. Some of the well-studied CI models include methylcholanthrene (MCA)-induced fibrosarcomas [69], ultraviolet B-induced skin cancer [70], azoxymethane/dextran sodium sulfate-induced colon carcinoma [71], 4-(methylnitrosoamino)-1-(3-pyridyl)-1-butanone (NNK) tobacco carcinogen-induced lung cancer [72,73]. In addition, many carcinogens such as N-methyl-N-nitrosourea (MNU) are used to create mouse, rat and other mammalian models to investigate antitumor therapy (discussed in [74]). Genome instability caused by a mutagen allows for an induction of de novo tumor formation in the corresponding microenvironment [75]. CI tumor models require a long time to establish, however, they have great genome complexity, which accurately reflects the real neogenesis of tumors in humans. Moreover, due to the greater mutational burden, different levels of immunogenic neoantigens are potentially generated, which affects the immunogenicity of the tumor [76]. Due to the potentially large number of unknown neoantigens, it is difficult to control and evaluate the immune response in CI models; therefore, they are rarely used to evaluate the effectiveness of CAR T-cell therapy or ICIs. However, the mouse model of 4-nitroquinoline 1-oxide (4NQO)-induced premalignant oral lesions that progress to oral cancer has been used to evaluate the effectiveness of the anti-PD-1 antibody [77,78].

## 5. Human Xenograft Models

The models described above allow the establishment of mouse tumors with a microenvironment consisting of mouse cells, which makes it possible to elaborately study the mechanisms of the tumor formation and their interactions with TME cells. However, to evaluate the effectiveness of immunotherapeutic approaches, models with human tumors that interact with cells of the human immune system are required, since the components of the mouse immune system do not always correspond to those in humans [79]. For this reason, a huge number of encouraging therapeutics, which showed promising results in mice in preclinical trials, were not able to show effectiveness in human trials [80]. Human xenograft models are immunocompromised hosts, into which human cells are transplanted. One of the oldest and most widespread animals for creating human xenografts is the athymic nude mice or severe combined immunodeficiency (SCID) mice, which are commonly used to evaluate the effectiveness of cytotoxic drugs [81]. Classic athymic nude mice have abnormal development of the thymus, which leads to severe T-cell dysfunction, however, they still have innate immune system components (neutrophils and dendritic cells (DCs), NK cells) and B cells [82]. SCID mice are deficient in DNA-dependent protein kinase, which is essential for the development of T- and B cells [83]. The engraftment efficiency of human tumors in SCID mice is higher than that compared to nude mice [84]. Also, SCID mice were the first to transplant human hematopoietic stem cells (HSCs) and peripheral blood mononuclear cells (PBMCs) [85,86]. However, the NK cell activity greatly limits the engraftment of human hematopoietic elements and other primary cells in such models; therefore, athymic nude or SCID mice are mainly used for the transplantation of human tumor cell lines [87]. However, it is worth noting that the engraftment rate of gastrointestinal tumors in nude mice is relatively high, while the engraftment of hematological malignancies is almost impossible [88]. Knockout of the *Rag1* and *Rag2*, as well as the interleukin (*IL2)rγ* genes made it possible to create of NOD SCID gamma (NSG) mice with impaired functions of IL2, IL4, IL7, IL9, IL15, and IL21 receptors and the absence of NK cells that allows successful transplantation of the primary tumors and human immune components [89]. However, NSG models and other immunocompromised models are also actively used for transplantation of tumor cell lines [90]. For example, CD19 CAR T-cell therapy controlled the progression of Raji B-cell lymphoma in the NSG model [91]. Adoptive transfer of glypican 3 (GPC3)-CAR T-cells suppressed growth of HepG2 and Huh-7 cell lines, which were subcutaneously transplanted in immunodeficient NOD/SCID/IL2rg−/− (NSI) mice [92].

### Patient-Derived Xenograft Models

Transplantation of primary patient tumors allows the creation of patient-derived xenograft models (PDX) that accurately reflect the complexity mediated by the natural development of the tumor, including genomic heterogeneity, tumor architecture and microenvironment factors, especially if the tumors are transplanted orthotopically [93,94]. PDX models of hematopoietic and lymphoid tissue tumors are actively used to evaluate and develop new CAR T-cell therapies aimed at various tumors [95,96]. The greatest success was achieved in the treatment of hematological tumors. For example, CAR T-cells targeting the thymic stromal lymphopoietin receptor (TSLPR) eliminated leukemia in 4 acute lymphoblastic leukemia xenograft models with overexpression of human cytokine receptor like factor 2 (CRLF2) [97]. The administration of human epidermal growth factor receptor 2 (HER2)-specific CAR T-cells led to regression or even elimination of the colorectal cancer xenograft and protection of relapse from rechallenged colon cancer tissue in the PDX model [95]. One of the promising areas that is being investigated in PDX models is the creation of CAR T-cells, the activation of which depends on several antigens in order to minimize side effects on healthy tissues. For example, the use of bispecific CAR T-cells targeting CD19 and CD20 allowed killing mixed populations of CD19 leukemia cells in the NSG model [96].

However, in order to effectively evaluate other types of immunotherapy, in particular ICIs, it is necessary to reconstruct the immune system, ideally from the same patient from whom the tumor was obtained. HSCs can be used to reconstruct the immune system (Figure 1D), both autologous and allogeneic HSCs can be used, since the collection of HSCs from a cancer patient may be limited [98]. Transplanted CD34^+^ HSCs develop into a complex human immune system without the risk of a graft-versus-host reaction (GVHR), since thymus education of human T-cells occurs in the context of mouse MHC molecules [99]. In immunocompromised mice, HSCs differentiate into helper T-cells, cytotoxic T-cells, B cells, monocytes, NK cells and DCs [100]. To date, the use of PDX models with HSC-reconstructed immune systems is the most common way to evaluate the treatment with anti-PD-1 and anti-CTLA4 antibodies [101,102]. For example, in the model of humanized cord blood (CB)-HSC mice, nivolumab (anti-PD-1 antibody) inhibited the growth of MDA-MB-231 triple negative breast cancer and CRC172 colorectal cancer cells, stimulating an antitumor T-cell response by the increase in the number of granzyme B (GrB) or interferon (IFN)-γ CD8^+^ T-cells in the tumors [102]. However, the use of allogeneic CD34^+^ HSCs in the same model can give different results. For example, it was shown that the therapy with pembrolizumab (anti-PD-1 antibody) both suppressed and did not affect the growth of bladder cancer and non-small-cell lung carcinoma (NSCLC) in NSG mice with the immune system reconstructed with HSCs from different donors. However, the authors did not find any connection between the effectiveness of the therapy with the level of mismatch of human leukocyte antigen (HLA) class I and II or with the number PD-1^+^ leukocytes. In addition, the evaluation of therapy effectiveness can be complicated by the fact that due to the partial mismatch of HLA between the tumor and immune cells, human T-cells can inhibit tumor growth regardless of therapy [101]. NSG mice transplanted with human CD34^+^ cells can be used to evaluate the toxicity of CAR T-cells against HSCs. For example, the use of CAR T-cells targeted at the CD44v6 adhesion receptor prevented the engraftment of acute myeloid leukemia and multiple human myeloma in the CD34^+^ humanized mice. The only side effect was transient monocytopenia, which cleared up after a decrease in the number of CAR T-cells [103].

The main difficulty PDX models may face in the context of immunotherapy adjustments for patients is the time it takes to produce the in vivo tumor model. For a reasonable time, it is necessary to successfully transplant a tumor into several animals, as well as reconstruct the patient’s hematopoietic system [104]. In order to quickly restore the autologous human immune system in immunocompromised mice with the patient’s tumors, PBMCs from adult donors can be used, since it is not necessary to wait for the formation of differentiated cells from HSCs up to 10–14 weeks [105,106]. PBMCs are also much easier to collect from patients and their engraftment time is reduced to four weeks [106]. However, the lifespan of PBMCs in PDX animals is much shorter compared to HSCs, so the time to evaluate immunotherapy is reduced to 4–8 weeks (Figure 1D) [86]. In addition, the administrated PBMCs generate stable GVHR [107]. In order to support the engraftment of transplanted cells, animals expressing proteins of the human immune system have been developed on the basis of NSG mouse models [98]. For example, NSG-SGM3 mice express human IL3, granulocyte macrophage colony stimulating factor (GM-CSF), and stem cell factor (SCF), which allows stable engraftment of human HSCs. This model promotes long-term engraftment with expanding populations of T-cells (CD4^+^, CD8^+^ and regulatory T-cells (Tregs)), B cells and myeloid cells [108]. Another MISTRG mouse model supports the engraftment and development of innate immune system cells, in particular myeloid cell, through the production of human signal regulatory protein α (SIRPα), macrophage colony-stimulating factor (M-CSF), IL3, GM-CSF and thyroperoxidase (TPO) [109,110]. The antitumor activity of nivolumab, atezolizumab (anti-PD-L1 antibody), pembrolizumab and other ICIs has been evaluated on PBMC-humanized mouse models [106,111]. In mice with gastric carcinoma and PBMCs from the same patient, co-administration of urelumab (human IgG4 monoclonal antibody, which targets 4-1BB) and nivolumab was sufficient to significantly slow tumor growth [111]. One study has indicated that a lung cancer model based on PBMCs and xenograft cell line derived from the same donor was more accurate when evaluating PD-L1/PD-1 immunotherapy, compared to an HSC-based model of the same tumor [106].

In highly immunocompromised mice, individual immune cell populations can also be restored. Functional human NK cells, Tregs and γδT-cells can be restored in immunodeficient animals and used to evaluate the antitumor effects of these cells or their effects on immunotherapy [112,113,114]. The use of mouse models with a partially restored immune system allows for evaluation of the effect of CAR T-cells on other components of the human immune system [114,115]. In one study, renal cell carcinoma and NK cell-transplanted NSG mice were treated with carbonic anhydrase IX (CAIX)-targeted CAR T-cells that secrete anti-PD-L1 antibodies. The secreted antibodies were able to recruit human NK cells to the tumor site by binding to the Fcγ receptor on the NK cell surface [115]. In the nude mouse model, it was demonstrated that CAR T-cells with CD28 co-stimulatory domain were ineffective against solid CEA^+^ tumors due to the high Treg tumor infiltration associated with the presence of the CD28 domain [114].

Comparative characteristics of various mouse tumor models are represented in Table 1.

## 6. Conclusions

Mice are good models for evaluating the effectiveness of immunotherapeutic agents because these laboratory animals are easy to maintain, relatively cheap, have a short lifespan, and short reproductive cycle. Syngeneic tumor models make it possible to create an in vivo model of a tumor by transplanting mouse tumor cell lines into immunocompetent animals in a short period of time. Such models are used for investigating the interaction of the tumor and the immune system, however, they do not reflect all the difficulties of both the tumor itself and its microenvironment. The rapid kinetics of tumor growth in syngeneic models often provides an inadequate time interval for evaluating the effectiveness of immunotherapy. In addition, to study the effectiveness of CAR T-cell therapy in syngeneic models, CAR T-cells have to be derived from murine cells, and the encouraging results of such studies often fail to repeat in clinical trials. Therefore, they cannot provide adequate models for the study of immunotherapeutic drugs for the treatment of human cancer.

Tumor GEMMs provide more opportunities for testing new immunotherapeutic agents due to de novo tumor formation and better TME modeling. The sequential development of the tumor allows the formation of more complex TME and provides a more complex interaction with the immune system, since the tumor passes through all stages of immunosurveillance. GEMMs are very important for investigating the undesirable side effects associated with CAR T-cell therapy, as this tumor model can be modified to express human TAAs, which are expressed not only in tumors, but also at lower levels in healthy tissues. However, since the creation of GEEMs is a very time-consuming and expensive process, and again the evaluation of immunotherapy in mouse tumors with TME represented by mouse cells yields mixed results, the use of these models to evaluate CAR T-cell and ICI immunotherapy is limited.

CI mouse models are usually not used to evaluate the effectiveness of CAR T-cell therapy or ICIs due to the fact that in such animals it is difficult to control and evaluate the immune response due to the large number of unknown neoantigens.

Transplantation of human tumor cell lines or primary tumors into immunocompromised mice allows the creation of human xenograft models, which are the most used method for the study of CAR T-cell therapy and ICIs. The reconstruction of the human immune system in xenograft models allows investigating aspects of human biology, such as the effect of various cytokines or other populations of human immune cells on the effectiveness of immunotherapy. The human immune system in animals can be reconstructed using CD34^+^ HSCs or PBMCs, which makes it possible to obtain the most accurate PDX models in which the tumor and immune components are obtained from the same donor. The development of this direction looks most promising, since such models can allow immunotherapy adjustments for a particular patient. These models may more accurately reflect a patient’s immune responses compared to current models with partial HLA matching. However, human bone marrow-derived stem cells have a limited ability to develop into immune cells in humanized mice, and PBMCs have limited engraftment possibilities. Therefore, mouse models that support the engraftment and development of human immune cells in humanized mice due to the secretion of cytokines and growth factors that do not cross-react between the mouse and human have been created. Improving these models will allow the formation of the fully functional human immune system in xenograft models and provide a significant tool for assessing the effectiveness of cancer immunotherapy.

Thus, today we have at our disposal mouse models, none of which can reflect all aspects of the complex genetics and biology of human cancer. Understanding the advantages and disadvantages of each model will ensure the most efficient use of the systems available today in order to facilitate the development of immunotherapeutic drugs and enhance continued evaluation of their effectiveness. At the same time, the development of new mouse models reflecting the interaction of the tumor and TME in as much detail as possible will increase the efficiency of translation of encouraging results of immunotherapy from bench to clinic.

## Figures and Tables

**Figure 1 ijms-21-04118-f001:**
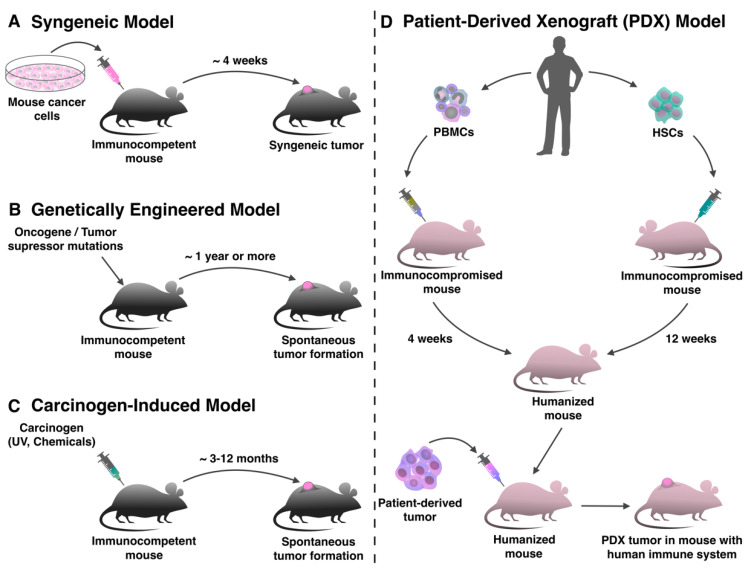
Production of mouse tumor models for evaluation of immunotherapy. (**A**) Syngeneic tumor models are produced by transplanting mouse tumor cell lines into immunocompetent animals in a short period of time. (**B**) In genetically engineered tumor models, the tumor forms de novo as a result of specific genome modification. (**C**) In mice treated with carcinogens, the tumor also forms spontaneously de novo. (**D**) For the production of patient-derived xenograft (PDX) models with human immune system immunocompromised mice can be humanized with peripheral blood mononuclear cells (PBMCs) or hematopoietic stem cells (HSCs), and after that transplanted with a human tumor.

**Table 1 ijms-21-04118-t001:** Comparative characteristics of mouse tumor models.

Tumor Model		The Origin of the Tumor	Heterogeneity of the Tumor	Complexity of the Production	Complexity of the Microenvironment	Complexity of the Immune System	Price
**Syngenic tumor models**		Transplanted mouse tumor cells	Low	Easy to set up, rapid tumor development	Tumor does not form a natural microenvironment	Fully functional mouse immune system	Low in cost
**Genetically engineered models**		*De novo* formed tumor induced by introduced mutations	Higher than in syngeneic models, depends on the production method	Difficult to set up, time-consuming	The tumor forms a natural microenvironment	Fully functional mouse immune system	High in cost
**Carcinogen** **-** **induced tumor models**		*De novo* formed tumor induced by carcinogens	High	Difficult to set up, time-consuming	The tumor forms a natural microenvironment	Fully functional mouse immune system	High in cost
**PDX models**	HSC-humanized	Patient-derived tumor	High	Difficult to set up, 10–12 weeks are required for HSC engraftment	TME is partially transplanted from the patient, but its complexity depends on the place of transplantation and the donor of the immune cells	The complex human immune system, no GVHR	High in cost
	PBMC-humanized	Patient-derived tumor	High	Difficult to set up, short time engraftment		The complex human immune system, induce GVHR	High in cost

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
