# Peer review of "Mouse Tumor Models for Advanced Cancer Immunotherapy"

_ijms, 2020, doi:10.3390/ijms21114118_

Round 1
Reviewer 1 Report
This is a very interesting review paper well planned and precisely devoted to a very important scientific problem.
My only recommendation is to involve the MNU-induced mammary carcinoma model in rats. For years it served as one of the best models of the disease in humans.
Author Response
Thank you for your comments and high scores for the manuscript. We agree that the use of rat models has made a major contribution to the study of antitumor therapy. However, since our article discusses mouse models specifically, we added brief information about MNU-induced tumor models in rats and linked it to a related study that comprehensively discuss this topic (Page 5, Line 182-184).
Reviewer 2 Report
Authors evaluated the role of the available mouse tumor model systems in relation to their effectiveness in assessing therapeutic benefits of immunotherapeutic agents, with particular focus on CAR T-cells and immune checkpoint inhibitors. They provided a review of the applicability and limitations of the various model systems, while also incorporating reference studies that highlight this topic.
Overall, manuscript is well written. However, the reading audience is better served and the review content more enriched if authors provide more extensive manuscript content on the subject matter. More reports of studies buttressing examples & illustrations of various cancer types mentioned in the different mouse model types may need be further provided.
Some more background information or heading about cancer immunotherapy ought to have been provided by authors within the manuscript.
In line 94, C7BL/6 is written instead of C57BL/6.
Author Response
Thank you for your comments which have helped us to improve our manuscript. We have corrected them point by point within the manuscript accordingly (your comments are in bold text and our responses are in ordinary type):
- More reports of studies buttressing examples & illustrations of various cancer types mentioned in the different mouse model types may need be further provided.
We have added some more examples of the use of various models (Page 3, Line 110-115; Page 4, Line 166-172; Page 6, Line 222-224; Page 7, Line 262-266).
- Some more background information or heading about cancer immunotherapy ought to have been provided by authors within the manuscript.
The appropriate information was added in the manuscript (Page 1, Lines 37-49).
- In line 94, C7BL/6 is written instead of C57BL/6.
It was corrected.